# An enzymatic activation of formaldehyde for nucleotide methylation

Charles Bou-Nader [1,6,7], Frederick W. Stull [2,7], Ludovic Pecqueur [1,7], Philippe Simon[1], Vincent Guérineau[3], Antoine Royant[4,5], Marc Fontecave [1], Murielle Lombard[1], Bruce A. Palfey[2] & Djemel Hamdane [1✉]

Folate enzyme cofactors and their derivatives have the unique ability to provide a single carbon unit at different oxidation levels for the de novo synthesis of amino-acids, purines, or thymidylate, an essential DNA nucleotide. How these cofactors mediate methylene transfer is not fully settled yet, particularly with regard to how the methylene is transferred to the methylene acceptor. Here, we uncovered that the bacterial thymidylate synthase ThyX, which relies on both folate and flavin for activity, can also use a formaldehyde-shunt to directly synthesize thymidylate. Combining biochemical, spectroscopic and anaerobic crystallographic analyses, we showed that formaldehyde reacts with the reduced flavin coenzyme to form a carbinolamine intermediate used by ThyX for dUMP methylation. The crystallographic structure of this intermediate reveals how ThyX activates formaldehyde and uses it, with the assistance of active site residues, to methylate dUMP. Our results reveal that carbinolamine species promote methylene transfer and suggest that the use of a $CH_2O$-shunt may be relevant in several other important folate-dependent reactions.

[1] Laboratoire de Chimie des Processus Biologiques, CNRS-UMR 8229, Collège De France, Université Pierre et Marie Curie, Paris, France. [2] Programs in Chemical Biology and the Department of Biological Chemistry, University of Michigan Medical School, Ann Arbor, MI, USA. [3] CNRS, Institut de Chimie des Substances Naturelles UPR 2301, Université Paris-Saclay, Gif-sur-Yvette, France. [4] CEA, CNRS, Institut de Biologie Structurale (IBS), Université Grenoble Alpes, Grenoble, France. [5] European Synchrotron Radiation Facility, Grenoble, France. [6] Present address: Laboratory of Molecular Biology, National Institute of Diabetes and Digestive and Kidney Diseases, Bethesda, MD 20892, USA. [7] These authors contributed equally: Charles Bou-Nader, Frederick W. Stull, Ludovic Pecqueur. ✉email: djemel.hamdane@college-de-france.fr

One-carbon transfer reactions are essential for the biosynthesis of many important metabolites including amino acids and nucleotides. Folate and methanopterin derivatives serve as central biological cofactors[1–7] due to their ability to provide a one-carbon unit at various oxidation levels[3]. Folate-dependent methylene transfer reactions are thought to proceed through the attack of an electrophilic iminium intermediate, derived from folate (Fig. 1) by a nucleophile, and thus yielding a methylene bridge between the cofactor and the carbon acceptor[5,8–11]. However, the mechanism of methylene transfer mediated by these folate cofactors is not fully understood[12,13]. The postulated iminium species have never been directly observed in any folate-dependent enzymes, while alternative mechanisms involving carbinolamine species or formaldehyde ($CH_2O$) as a reaction intermediate have been considered[9,12,14], but remained difficult to demonstrate (Fig. 1).

The reductive methylation of deoxyuridylate (dUMP) into deoxythymidylate (dTMP) is a critical step in DNA biosynthesis. In prokaryotes and some archaea, the homotetrameric flavin-dependent thymidylate synthase homotetrameric ThyX catalyzes dUMP methylation into dTMP, making ThyX mandatory for cell survival in the absence of external sources of thymidylate[15]. ThyX relies on the flavin adenine dinucleotide (FAD) as well as $N^5,N^{10}$-methylenetetrahydrofolate ($CH_2THF$) as methylene donor for activity, while nicotinamide adenine dinucleotide phosphate (NADPH) acts as an electron donor. This enzyme forms a distinct class of thymidylate synthase that differs from the human thymidylate synthase TYMS or prokaryotic ThyA in terms of structure and mechanism. Indeed, the homodimeric ThyA uses $CH_2THF$ for both the one-carbon methylene and the reducing hydride to form the C7 methyl of the dTMP product[16]. Since ThyX is found nearly exclusively in prokaryotes (with *Dictyostelium* being the exception), especially in severe pathogens[15,17,18], it represents a promising antimicrobial target.

The mechanism of ThyX has been debated and revised many times since the enzyme's discovery and is still unsettled[19–25]. The carbon transfer reaction was initially thought to proceed via the direct transfer of a $CH_2$ from the folate to the dUMP substrate. However, such a mechanism became untenable after the folate binding site was identified in crystal structures[26], which showed that the FAD cofactor binds between the folate and dUMP in the ternary complex (Supplementary Fig. 1) and that the folate cannot react directly with dUMP. An alternative mechanism suggested that the $CH_2$ is first transferred to an arginine residue present in the active site prior to its transfer to the dUMP[21]. However, such a mechanism was discarded since the mutation of this arginine did not abolish ThyX's activity[26]. Recently, the detection of a reaction intermediate by chemical quenching, assigned to be derived from a covalent $FAD-CH_2-dUMP$ adduct[27], suggested that the flavin acts as a methylene transfer agent between $CH_2THF$ and the substrate, presumably through an iminium species. Here, we provide evidence for a flavin-carbinolamine species as the relevant methylene donor for ThyX

catalysis. Combining mass spectrometry (MS), nuclear magnetic resonance (NMR), and kinetic analyses, we show that $CH_2O$ can replace the natural methylene donor for ThyX-dependent dUMP methylation. Moreover, we uncover the molecular details of a flavin-carbinolamine species in ThyX active site by anaerobic crystallography, expanding the repertoire of N5-alkylated flavins used for biocatalysis.

## Results

**ThyX uses $CH_2O$ as a direct methylene donor.** We considered the possibility of using a $CH_2O$-shunt reaction in the catalytic cycle of ThyX. In such a scenario, FAD in its two-electron reduced $FADH^-$ form could activate $CH_2O$ leading to a flavin-carbinolamine species, which should naturally be in equilibrium with its iminium counterpart[28] (Supplementary Fig. 2). This alternate route to generate a reactive enzyme intermediate is conceptually analogous to the hydrogen peroxide shunt employed to bypass the two-electron transfer and oxygen-binding steps in some artificial flavin, heme, and non-heme-dependent hydroxylases catalytic cycle[29]. To substantiate this hypothesis, we tested under anaerobic conditions whether recombinant *Thermotoga maritima* ThyX methylates dUMP in the presence of NADPH and $CH_2O$, but in the absence of $CH_2THF$. Analysis of the products by mass spectrometry (MS) confirmed the formation of dTMP only in the simultaneous presence of ThyX, NADPH, and $CH_2O$ (Supplementary Fig. 3). The analysis was repeated with $^{13}C$-labeled $CH_2O$ and showed that the methylene incorporated in dTMP did not come from sources other than $CH_2O$ since only [$^{13}C7$]-dTMP was formed, as unambiguously detected by nuclear magnetic resonance (NMR) spectroscopy and MS analyses (Fig. 2 and Supplementary Figs. 3 and 4). Thus, ThyX is responsible for the $CH_2O$-dependent dUMP methylation.

Since NADPH was strictly required for dTMP formation (Supplementary Fig. 3), this suggested that an electron donor was needed consistent with a reductive methylation. To confirm this, we observed that the addition of $CH_2O$ to the pre-reduced ThyX $FADH^-·dUMP$ complex, in anaerobic conditions and in the absence of NADPH, resulted in flavin oxidation and dTMP production (Supplementary Fig. 5). This establishes that the measured $CH_2O$-dependent ThyX activity is a reductive methylation. Indeed, stopped-flow experiments mixing ThyX $FADH^-·dUMP$ with $CH_2O$ under anaerobic conditions revealed the transient formation of a flavin intermediate followed by a biphasic flavin oxidation process (Fig. 2b). The observed rate constant for the formation of this intermediate increases linearly with $CH_2O$ concentration, consistent with a reversible bimolecular reaction ($k_{on} = 1.1 \pm 0.01$ $M^{-1}$ $s^{-1}$, $k_{off} = 0.022 \pm 0.003$ $s^{-1}$, $K_D$ of ~$20 \pm 3$ mM), while rate constants for flavin oxidation showed hyperbolic dependences ($K_D = 23 \pm 8$ mM, $k_{ox} = 0.03 \pm 0.001$ $s^{-1}$ and $K_D = 29 \pm 10$ mM, $k_{ox} = 0.01 \pm 0.001$ $s^{-1}$) (Fig. 2c). Taken together, these results confirmed that ThyX uses $CH_2O$ as a direct methylene donor for dTMP synthesis and can therefore

**Fig. 1 Hydrolysis reaction of $N^5,N^{10}$-methylenetetrahydrofolate as a source of formaldehyde.** The cyclic form of $CH_2THF$ is activated by protonation leading to the iminium species ($CH_2THF^+$). This iminium is potentially in equilibrium with its carbinolamine counterpart ($HOCH_2THF$). The carbinolamine intermediate can readily decompose into THF and formaldehyde ($CH_2O$) in a reversible reaction[28]. $CH_2O$ can then serve as a direct source of methylene.

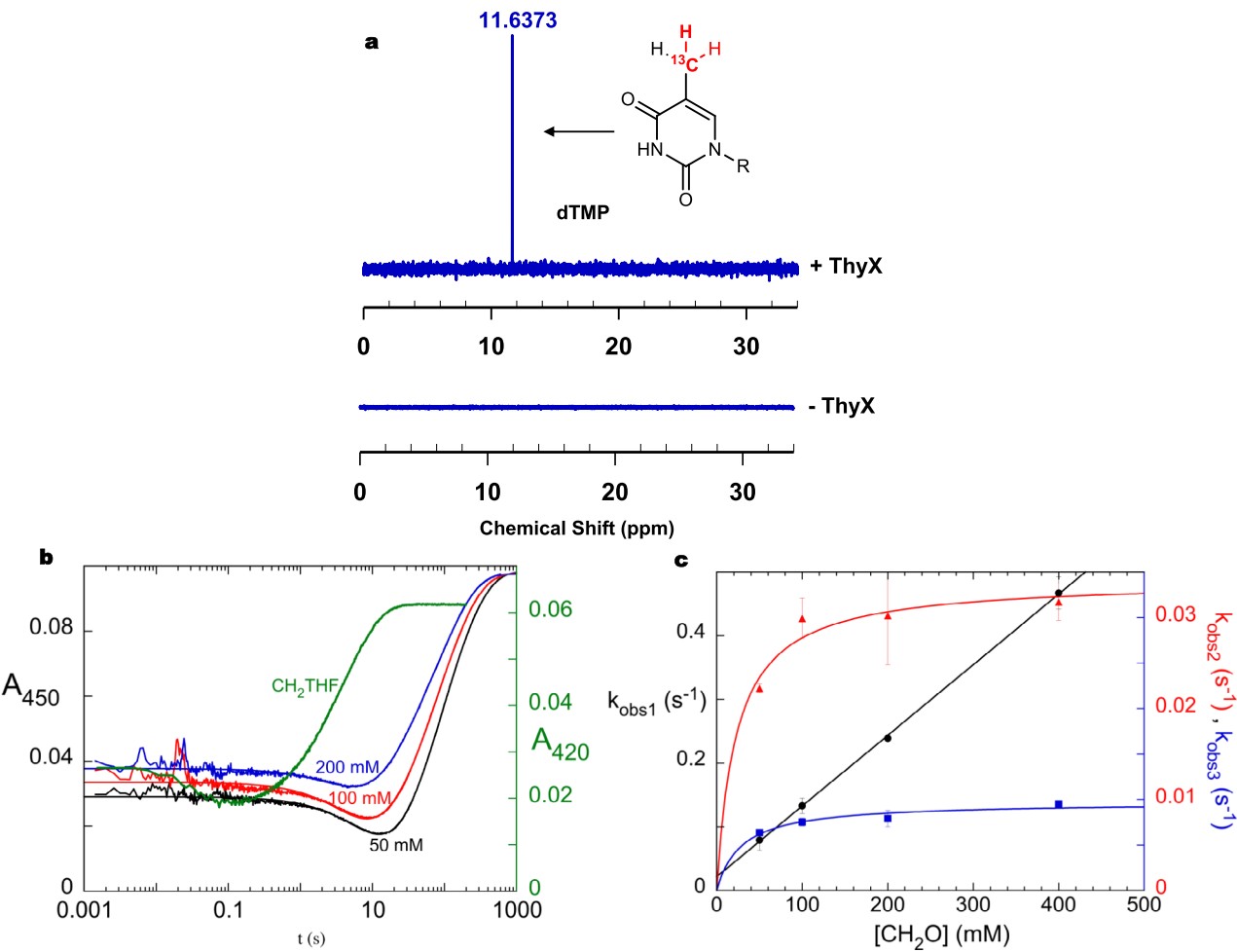

**Fig. 2 Use of formaldehyde as a direct methylene donor in the *T. maritima* ThyX-catalyzed dUMP methylation. a** $^{13}$C-NMR spectrum of dTMP formed by ThyX in the presence of NADPH, dUMP, and $^{13}$C-labeled $CH_2O$ (top). At the bottom is shown the $^{13}$C-NMR spectrum of the control experiment conducted in the presence of NADPH, dUMP, and $^{13}$C-labeled $CH_2O$ without adding ThyX. **b** Reaction of ThyX with formaldehyde. The reaction kinetics were investigated at different formaldehyde concentrations in stopped-flow experiments. The reduced enzyme-dUMP (FADH⁻•dUMP) complex was loaded into a stopped-flow instrument and mixed rapidly with solutions containing high concentrations of formaldehyde. Three examples of absorbance traces are shown using a logarithmic timescale. Traces were fit to a sum of three exponentials. For comparison, the kinetic in the presence of the physiological methylene donor, $CH_2THF$, is shown in green. **c** The observed rate constants of each phase varied with formaldehyde concentration. The observed rate constant ($k_{obs}$) of the fastest phase (black), describing the decrease in absorbance at 450 nm, increased linearly with formaldehyde concentration, consistent with a reversible bimolecular reaction of enzyme and formaldehyde: $1.11 \pm 0.01 M^{-1} s^{-1}$, $0.022 \pm 0.003 s^{-1}$, giving a $K_D$ of $20 \pm 3$ mM. The $k_{obs}$ values for the two subsequent phases (red and blue data points) increased hyperbolically with formaldehyde concentration: maximum $k_{obs2}$ of $0.034 \pm 0.001 s^{-1}$, $K_D$ of $23 \pm 8$ mM and maximum $k_{obs3}$ of $0.0097 \pm 0.0006 s^{-1}$, $K_D$ of $29 \pm 10$ mM. The error bars are the standard deviation calculated from three independent experiments.

substitute for $CH_2THF$ by forming a flavin intermediate that could be a carbinolamine adduct.

**A flavin carbinolamine sustains ThyX activity.** One way to firmly establish that activation of $CH_2O$ by FADH⁻ proceeds via a flavin carbinolamine would be to activate an apoprotein version of ThyX (apo-ThyX) with a synthetic flavin carbinolamine compound if the latter is stable enough to be used. Earlier studies reported that 1,5-dihydrolumiflavin derivatives, used as chemical models of reduced flavin, readily react with $CH_2O$ to form an N5 carbinolamine adduct easily identifiable by its light absorption features[30]. However, this has never been tested directly with the natural coenzyme. We found that the addition of $CH_2O$ to an anaerobic solution of FADH⁻ produces a stable flavin adduct tentatively assigned to the synthetic flavin carbinolamine on the basis of its optical spectrum (Supplementary Fig. 6A). This synthetic flavin carbinolamine binds to apo-ThyX and is consumed

upon anaerobic addition of dUMP leading to dTMP and oxidized FAD (Supplementary Fig. 6B), showing that flavin carbinolamine is a dUMP-methylating agent. This provides an example of a synthetic compound mimicking a potential key intermediate being active for dUMP methylation. We had previously used a similar strategy to activate a flavin- and folate-dependent transfer RNA (tRNA) methyltransferase, TrmFO, using a synthetic compound mimicking the tRNA-methylating agent, namely, a FAD-$CH_2$-cysteine adduct observed in the freshly purified enzyme[11,31,32].

**Structural capture of flavin carbinolamine in ThyX active site.** We used crystallography under anaerobic conditions to provide further molecular insights into $CH_2O$ activation by ThyX, taking advantage of the fact that flavin carbinolamine is a relatively long-lived compound under anaerobic conditions. Well-diffracting crystals of the apo-ThyX/synthetic flavin carbinolamine complex

were obtained in a glove box. In addition, we attempted to study the reaction of reduced ThyX with $CH_2O$ in crystallo. For that purpose, we crystallized reduced ThyX in a glove box and then soaked the resulting crystals with $CH_2O$ in the absence of dUMP under anaerobic conditions (FADH$^-$•$CH_2O$). Under these conditions, we found that $CH_2O$ diffused within ThyX active site and reacted with FADH$^-$ in crystallo, thus forming a flavin carbinolamine as ascertained by the optical spectrum of (FADH$^-$•$CH_2O$) crystal recorded on a micro-spectrophotometer directly coupled at the X-ray beamline at the synchrotron (Supplementary Fig. 7). Structures of apo-ThyX/synthetic flavin carbinolamine and FADH$^-$•$CH_2O$ were determined at 2.8 and 2 Å resolution, respectively (Supplementary Table 1). The structures show that the homotetrameric enzyme does not undergo significant conformational changes with an overall root-mean-square deviation of 0.2 Å over 167 residues (Supplementary Fig. 8).

A clear additional density on FAD was observed in both structures, specifically in three out of four FAD molecules present in apo-ThyX/synthetic flavin carbinolamine and in two FAD molecules in FADH$^-$•$CH_2O$ (Fig. 3a, b and Supplementary Figs. 9–11). This density fits with a $CH_2OH$ group attached to the N5 atom of the isoalloxazine ring and is thus attributed to the N5-C5a carbinolamine FAD adduct in both structures: in the first case, it is the synthetic compound, while in the second it is the product of the reaction between FADH$^-$ and $CH_2O$ in crystallo. The carbinolamine adduct adopts a butterfly conformation, significantly bent along its C10a–C4a axis with a dihedral angle

~12° and its N5 is pyramidal, consistent with an $sp^3$ hybridized nitrogen and alkylation of N5. In both structures, the carbinolamine groups adopt a similar axial position lying toward the *si*-face of FAD and interact with a water molecule. In the case of the FADH$^-$•$CH_2O$ structure, an inorganic phosphate occupies the dUMP-binding site and makes an additional interaction with the carbinolamine moiety (Fig. 3a).

**Flavin carbinolamine as a bona fide methylene donor.** Our structures are perfectly superimposable with that of the ThyX/dUMP/folate ternary complex previously reported, with the phosphate ligand placed at a position similar to that of the phosphate group of dUMP. This allows us to provide a model of a catalytically relevant species in which the flavin carbinolamine is sandwiched between dUMP and folate (Fig. 3c). In this model structure, the methylene group of the carbinolamine adduct is located only 2.4 Å from C5-dUMP and is ideally poised for $S_N2$ attack by the activated nucleobase to generate an intermolecular C–C bond, producing the FAD-$CH_2$-dUMP adduct (Supplementary Fig. 12). However, a prior reorientation of the β-hydroxyl leaving group would be mandatory to facilitate its departure during catalysis. For example, a simple clockwise or anti-clockwise rotation of this hydroxyl group would be sufficient to orient it optimally for the $S_N2$ attack (Fig. 3d). The structure shows that the strictly conserved Tyr91 and Ser88 (Fig. 3d and Supplementary Fig. 13), previously shown to be critical for

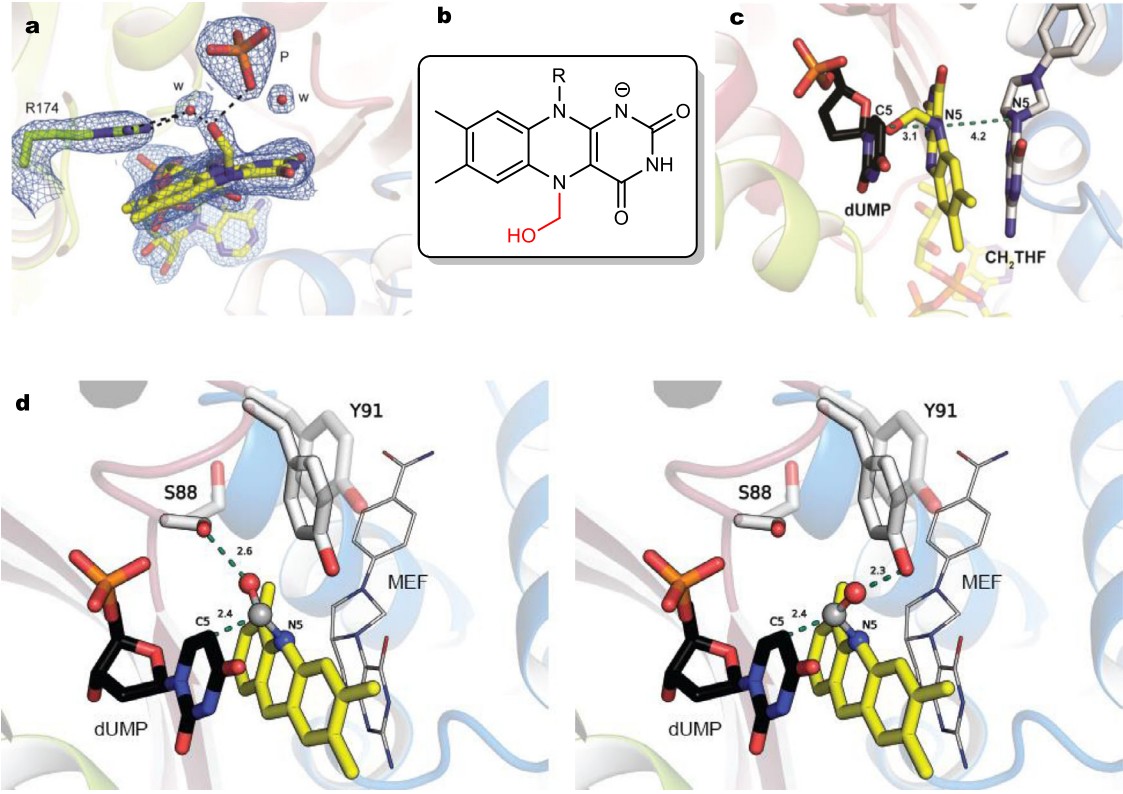

**Fig. 3 Activation of $CH_2O$ by *T. maritima* ThyX and structural capture of the carbinolamine flavin intermediate. a** Section of the 2mF0-DFc electron density contoured at 1σ around the flavin carbinolamine of ThyX obtained by the anaerobic reaction of $CH_2O$ with FADH$^-$ (FADH$^-$•$CH_2O$). Waters and a phosphate molecule are labeled w and P, respectively. R174 participates in dUMP activation[24] and interacts with an active site water molecule. **b** Chemical structure of the N5 flavin-carbinolamine adduct. **c** Model of ThyX in complex with the flavin carbinolamine, dUMP, and folate. The model was obtained by superimposing the crystal structure of FADH$^-$•$CH_2O$ complex with the structure of ThyX in complex with dUMP and folate (pdb, 4gt9). **d** Structural model of ThyX in complex with the flavin carbinolamine, dUMP, and folate, in which the carbinolamine is properly oriented for attack by the activated C5-dUMP. This SN2 is likely assisted by two residues, S88 and Y91, which are conserved among ThyX enzymes. This model shows the different orientations adopted by these residues in two different structures of ThyX and they may assist the SN2 and departure of the β-hydroxyl leaving group of the flavin carbinolamine. The structures used for this model are: (i) ThyX FADH$^-$ soaked with 20 mM $CH_2O$ vs (ii) pdb 4gt9.

**Fig. 4 Proposed chemical mechanism for ThyX.** FAD is first reduced by NADPH. Then, the reduced flavin, FADH⁻, reacts with the CH₂THF to form a carbinolamine flavin, which acts as the *genuine* methylene donor. The flavin carbinolamine can be obtained directly via a CH₂O-shunt reaction consisting of a reaction of FADH⁻ with free CH₂O. Methylene transfer from FAD to dUMP is initiated by an SN2 reaction of activated dUMP and the flavin carbinolamine, leading to water elimination and formation of a transient FAD-CH₂-dUMP adduct.

activity by site-directed mutagenesis[24], could stabilize such a conformation through hydrogen bonds (Fig. 3d). Both residues are dynamic and can adopt different conformations according to the presence of the carbinolamine adduct (Fig. 3d).

Based on the above data, we propose a revised mechanism for ThyX-dependent methylation of dUMP that uses a flavin carbinolamine as the key methylene donor as opposed to a flavin-iminium intermediate (Fig. 4). The reaction between the reduced flavin cofactor and CH₂THF results in the flavin carbinolamine and tetrahydrofolate (THF). The carbinolamine geometry favors an $S_N2$-like attack by dUMP, which is activated as a nucleophile by deprotonation of N3 through electrostatic interaction with R174, enhancing the enamine character of the pyrimidine moiety[33]. Nucleophilic attack leads to the formation of a covalent FAD-CH₂-dUMP adduct, previously proposed based upon the fragments obtained from rapid-quenching by a base[27], and the displacement of a water molecule. The two conserved residues, Ser88 and Tyr91, at hydrogen-bonding distance with the carbinolamine hydroxyl group may assist the $S_N2$ reaction by acting as acids and promoting the displacement of the β-hydroxyl leaving group (Fig. 3d). It is worth noting that the phenolic oxygen of Tyr91 is within hydrogen-bonding distance to a guanidinium nitrogen of Arg 90 (also conserved; Supplementary Fig. 13); the proximity of such a (presumably) positive charge could enhance the acidity of Tyr91, which might be the general acid catalyst that assists the displacement of the leaving water. This rationalizes our previous kinetic observations that *T. maritima* ThyX S88A, R90A, or Y91A mutants induced a large decrease (by more than two orders of magnitude for Y91A) in the rate constant for the consumption of the flavin carbinolamine triggered by dUMP[24].

Lastly, elimination of FADH⁻ results in the formation of a nucleotide with an exocyclic methylene group, which is subsequently reduced by FADH⁻ into thymidine. The cycle closes with the reduction of FAD by NADPH.

## Discussion

Our study presents the first example of the direct use of CH₂O as methylene donor instead of CH₂THF during an enzymatic methylation reaction, namely, thymidylate formation catalyzed by ThyX. The tighter binding of ThyX to CH₂THF ($K_D$ ~4 μM[34]) compared to CH₂O ($K_D$ ~20 mM) confirms that methylene tetrahydrofolate acts as the biological carbon donor for ThyX, serving as a CH₂O carrier/transporter and thus avoiding genotoxic effects[35–37]. This CH₂O shunt allowed us to isolate a catalytically active FAD derivative and to structurally characterize it as an N5 carbinolamine adduct bound in the active site of ThyX. This expands the known structural examples of an in crystallo capture of N5-alkylated flavin as a reactive enzyme species following the recently reported flavin-isopentenyl adduct found in a class of flavin-dependent decarboxylases and the galactopyranose linked to the FAD through a covalent bond between the anomeric C of galactopyranose and N5 of the FAD in UDP-galactopyranose mutase[38–40].

We propose here a revised mechanism for ThyX in which this flavin-carbinolamine species is the methylene donor via an acid-catalyzed $S_N2$ process that releases water and forms methylene-dUMP (Fig. 4). As this carbinolamine is theoretically in equilibrium with the corresponding flavin-iminium species (Supplementary Fig. 2), the latter could, as previously proposed, conceivably be the actual electrophile. However, this scenario is unlikely for the following reasons. First, the flavin-iminium is highly unstable and has

been shown to react with water rapidly to form the corresponding flavin carbinolamine[41]. Second, efficient attack of nucleophiles on the π-system of carbonyls or imines occurs along the so-called Bürgi–Dunitz trajectory[42], with the nucleophile attacking the unsaturated carbon with an obtuse angle of ~107° with respect to the C–X bond (X being the leaving group)[43]. In contrast, the $sp3$ hybridized carbinolamine presents a favorable distance and geometry for the in-line attack of the C5 carbon of dUMP for C–C bond formation and water displacement. Therefore, these stereoelectronic considerations lead us to favor the carbinolamine as the actual carbon transfer agent. Carbinolamine intermediates may also be present in other methylene transfer reactions (e.g., methylene transfer from $CH_2THF$ to dUMP in ThyA). Interestingly, a carbinolamine derivative of $CH_2THF$ formed in crystals of a C-terminal deletion mutant of ThyA with 5-fluoro-dUMP and $CH_2THF$ as added ligands[44]. This carbinolamine was dismissed as a side product of the postulated iminium intermediate. However, given our observations with ThyX, this folate-carbinolamine intermediate in ThyA may be the actual methylene transfer agent for this enzyme.

Overall, we generated a flavin-carbinolamine adduct in ThyX using a $CH_2O$ shunt that is harnessed to methylate dUMP and allowed us to shed light on the complex mechanism of ThyX involving the nucleophilic reactivity of the flavin coenzyme (Fig. 4). We anticipate that this finding will inform the design of molecules to target ThyX and offer new perspectives on mechanistic studies of carbon transfer in folate enzymology.

## Methods

**Protein expression and purification.** ThyX from *T. maritima* was expressed in a pET11d transformed in BL21(DE3) using LB medium following induction with 0.5 mM isopropyl β-D-1-thiogalactopyranoside at $OD_{600}$ of 0.6. After overnight incubation at 29 °C, cells were harvested by centrifugation and lysed by sonication in 50 mM sodium phosphate pH 8, 2 M NaCl, 10% glycerol, and 15 mM imidazole (buffer A) containing 10 mM β-mercaptoethanol and 2 mM phenylmethanesulfonyl fluoride. The lysate was centrifuged at $85,000 \times g$ for 60 min and the supernatant was heated for 10 min at 65 °C, followed by centrifugation at $235,000 \times g$ for 60 min to remove precipitated proteins and nucleic acids. The supernatant was then loaded on a 5 mL $Ni^{2+}$-NTA agarose affinity column (Qiagen) pre-equilibrated with buffer A and washed with 100 mL of buffer A. Bound ThyX was then eluted with 250 mM imidazole. The eluted protein was loaded onto a S200-Superdex GE gel filtration column equilibrated in 50 mM Tris-HCl pH 8 and 150 mM NaCl. Protein purity was checked by sodium dodecyl sulfate-polyacrylamide gel electrophoresis. To produce apo-ThyX, the protein was unfolded by washing extensively with 8 M urea on the $Ni^{2+}$-NTA affinity column and then eluted with 50 mM sodium phosphate pH 8 and 200 mM imidazole, followed by overnight dialysis in 8 M urea at 4 °C. Complete removal of cofactors was confirmed by ultraviolet (UV) absorption and apo-ThyX was refolded by overnight dialysis against 50 mM sodium phosphate pH 8, 100 mM L-arginine and 1 mM ethylenediaminetetraacetic acid at 4 °C.

**ThyX activity tests.** Twenty micromoles of 2′-deoxyuridine 5′-monophosphate (dUMP), 1 mM $CH_2O$ or $^{13}C$-labeled $CH_2O$, and 1 mM NADPH were incubated for 30 min in anaerobic conditions at room temperature with 50 μM of ThyX in 100 μL of 50 mM Tris pH 7.4 and 150 mM NaCl. The reaction was stopped by adding 100 μL of acidic phenol. Nucleotides in the aqueous phase were recovered after centrifugation and analyzed by high-performance liquid chromatography or NMR. Matrix-assisted laser desorption/ionization MS analyses were performed directly on the aqueous phase after phenol treatment of the samples using an UltrafleXtreme spectrometer (Bruker Daltonique, France). The instrument is equipped with an Nd:YAG laser (operating at 355 nm wavelength of <500 ps pulse and 200 Hz repetition rate). Acquisitions were performed in a negative ion mode. MS data were processed using DataExplorer 4.4 (Applied Biosystems).

**Reconstitution of apo-ThyX with synthetic flavin carbinolamine.** Flavin carbinolamine was synthesized in anaerobic conditions in a glove box by reducing FAD with 2 molar equivalents of dithionite freshly prepared, followed by the addition of 100 molar equivalent of formaldehyde ($CH_2O$). The mixture was buffered at pH 8 with sodium phosphate and left in the glove box at ~20 °C overnight. Four milligrams of apo-ThyX was incubated in 2.5 mL of 50 mM sodium phosphate pH 8 with the freshly prepared flavin mixture for 10 min under anaerobic condition. To remove excess flavin, reconstituted ThyX was further purified on a desalting column Pd10 (GE Healthcare) equilibrated in 50 mM Tris-HCl pH 7.4 and 150 mM NaCl.

**Reaction of reduced ThyX-FAD-dUMP ternary complex with formaldehyde.** A solution of oxidized ThyX-FAD-dUMP (9 μM tetramer) was made anaerobic in a sealed cuvette by repeated cycles of evacuation and equilibration with purified argon. ThyX was reduced with 1 molar equivalent of freshly prepared dithionite and mixed with anaerobic formaldehyde. The reaction was monitored in a Shimadzu UV-2501PC scanning spectrophotometer.

**Stop-flow experiments between $CH_2O$ and reduced ThyX-dUMP complex.** All stopped-flow experiments were performed in 0.1 M Tris-HCl (pH 8) at 25 °C using a TgK Scientific SF-61DX2 KinetAsyst stopped-flow instrument that had been previously equilibrated with a glucose/glucose oxidase solution to make the internal components of the system anaerobic. A solution containing ThyX (4 μM tetramer) and dUMP was made anaerobic in a glass tonometer by cycling with vacuum and argon. The ThyX-dUMP complex was stoichiometrically reduced by titration with dithionite, loaded onto the instrument, and mixed with buffer containing formaldehyde that had been bubbled with argon to make the solution anaerobic. Reaction traces were monitored at 450 nm and were fit to sums of exponentials using KaleidaGraph (Synergy Software) to obtain observed rate constants. The plot for $k_{obs1}$ against $CH_2O$ concentration was fit to a line to determine $k_{on}$ and $k_{off}$. The plots for $k_{obs2}$ and $k_{obs3}$ were each fit to a square hyperbolae to determine the apparent $K_D$ and $k_{ox}$ for these two phases.

**Crystallization of ThyX.** All crystals were obtained by vapor diffusion in a glove box. For crystallization of apo-ThyX/synthetic 4′, the apoenzyme was freshly reconstituted with 6′ in anaerobic conditions and concentrated up to 10 mg/mL. One microliter of the holoenzyme in 50 mM Tris-HCl pH 7.5 and 150 mM NaCl was mixed with 1 μL of reservoir composed of 40% polyethylene glycol 200 (PEG 200). Colorless crystals were obtained overnight and directly frozen in liquid propane. For crystallization of 5.CH₂O, freshly reconstituted ThyX with FAD was reduced with 2 molar equivalent of dithionite and further purified by a desalting PD10 column in 50 mM Tris-HCl pH 7.5 and 150 mM NaCl. One microliter of 7 mg/mL reduced ThyX was mixed with 1 μL of reservoir solution composed of 40% PEG 200. Colorless crystals grew overnight and were soaked for 30 min in 20 mM formaldehyde in 40% PEG 200 before flash freezing in liquid propane. All diffraction data were collected on single crystals at the microfocused PROXIMA-2 beamline at the SOLEIL synchrotron (Saint-Aubin, France) at 100 K using an Eiger X-9M. Data were indexed, processed, and scaled using XDS[45]. Both structures were solved by molecular replacement using a monomer from PDB 4gt9 as a search template and further refined with autoBUSTER[46]. A clear density for the FAD moieties was observed in both structures with additional density on the N5 modeled as $CH_2OH$ adduct. Chemical restraints for the carbinolamine flavin adduct were generated using JLigand and grade[47]. The UV–visible absorption spectrum of a crystal of ThyX in complex with the flavin carbinolamine compound was recorded at 100 K at the icOS Lab located at the ESRF in Grenoble[48].

**Reporting summary.** Further information on research design is available in the Nature Research Reporting Summary linked to this article.

## Data availability
The raw kinetic traces used in this study are provided in the Source data file. The atomic coordinates and structure factor amplitudes for the crystal structures of ThyX FADH⁻ soaked with $CH_2O$ and apo-ThyX/synthetic compound 4′ have been deposited in the Protein Data Bank (https://www.rcsb.org) under accession codes 7NDW and 7NDZ, respectively. Source data are provided with this paper.

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

## Acknowledgements

This work was supported by the Centre National de la Recherche Scientifique, Université Pierre et Marie Curie as well as the "Initiative d'Excellence" program from the French State (ANR-11-IDEX-0004-02, ANR-15-CE11-0004-01, Grant "DYNAMO," ANR-11-LABX-0011-01), and the United States National Science Foundation (CHE 1905267). This work used the icOS platform of the Grenoble Instruct-ERIC Center (ISBG; UAR 3518 CNRS-CEA-UGA-EMBL) within the Grenoble Partnership for Structural Biology (PSB), supported by FRISBI (ANR-10-INBS-0005-02) and GRAL, financed within the University Grenoble Alpes graduate school (Ecoles Universitaires de Recherche) CBH-EUR-GS (ANR-17-EURE-0003). We also acknowledge SOLEIL for the provision of synchrotron radiation facilities (proposal ID 20170872).

## Author contributions

B.A.P. and D.H. designed the work. C.B.-N. synthesized the flavin carbinolamine, purified ThyX and apo-ThyX, and established anaerobic crystallization conditions. C.B.-N. and L.P. solved the structures. C.B.-N, L.P., and M.L. purified proteins and prepared crystals. F.W.S. and B.A.P. performed stopped-flow kinetics. P.S. and M.L carried the activity tests and NMR experiments. V.G. performed the MS analysis. A.R. recorded UV-absorption spectra on crystals. D.H. wrote the initial draft. C.B.-N., F.W.S, L.P., P.S., V.G., A.R., M.F., B.A.P., and D.H. contributed to interpreting the data and writing the paper.

## Competing interests

The authors declare no competing interests.
