## [Peer Review File · Nature Communications]

REVIEWER COMMENTS

Reviewer #1 (Remarks to the Author):

This manuscript proposed the chemical mechanism of the methylation by ThyX to produce the thymidine nucleotide. Although the tertiary structures of ThyX from several organisms have been determined so far, the reaction mechanism has been unclear. The authors demonstrated the presence of the carbinolamine intermediate by the X-ray crystallography. Also, the authors showed that formaldehyde can be used instead of methylene-THF to produce dTMP from dUMP. Based on these results, a novel chemical mechanism for ThyX is proposed. The elucidation of chemical mechanism is fundamental in biochemistry and this manuscript gave new insight of the enzymatic reaction. However, the following points will be addressed by the authors to justify the proposed mechanism.

1. In the proposed chemical mechanism by Mishanina et al. 2016, the methylene group was transferred to flavin to form iminium intermediate (CH₂-flavin) instead of carbinolamine intermediate (CHO-flavin). Indeed, the S_N2 reaction between the C5 of dUMP and the methylene carbon of carbinolamine intermediate is reasonable. Still, it is possible that the carbinolamine intermediate is formed via iminium intermediate. Furthermore, it is also possible that iminium form is the active intermediate and the iminium intermediate is formed from carbinolamine intermediate in the CH₂O-shunt pathway. It is recommended to discuss on these possibilities.

2. Catalytic mechanism is also important. Because crystal structures were reported in the manuscript, the authors may discuss on the contribution of amino acid residues including Tyr91 and Ser88 to the catalytic mechanism of ThyX. For example, how the H₂O molecule is generated during the reaction? Furthermore, it is helpful for general readers to show the conservation of these amino acid residues in sequences as well as structures.

Reviewer #2 (Remarks to the Author):

Authors demonstrate that in anaerobic conditions formaldehyde can replace the natural methylene donor CH₂THF in ThyX catalyzed methylation of dUMP. These findings are highly interesting and endorsed by crystallographic, NMR and HPLC data. However, comparison of reaction kinetics with the native methylene donor should be included.

In the crystal structure a the carbinolamine adduct of reduced FADH₂ is observed. Authors are convinced this is the methylating agent and not a trapped intermediate that should be activated into a reactive imine since 'The methylene group of the carbinolamine adduct is nicely located only 2.4 Å from C5-dUMP and is ideally poised for an S_N2 attack by the activated nucleobase to generate an intermolecular C-C bond.' However, without MD these are not convincing arguments. An MD is the least that should be performed on the obtained model to have a more reliable idea on distances and hydrogen bonding interactions indicated in fig 3d, to endorse the hypothesis that carbinolamine flavin species could be the active species instead of its iminium counterpart, that is so far postulated as the dUMP methylating agent.

For comparison: To date, the 5-HOCH₂H₄folate that was considered as structural evidence for the 5-iminium ion intermediate, which is the proposed reactive form of CH₂H₄folate in ThyA (Biochemistry. 1993 Jul 20;32(28):7116-25. doi: 10.1021/bi00079a007) and also the 5-hydroxymethyl-dUMP was identified as an acid trapped intermediate in the reaction catalyzed by ThyX (T.V. Mishanina, et al. Trapping of an intermediate in the reaction catalyzed by flavin-dependent thymidylate synthase J. Am. Chem. Soc., 134 (9) (2012), pp. 4442-4448).

The top spectrum in fig 2A is an expansion of the full NMR spectrum (not RMN!) in Fig S4C. The full spectrum clearly shows many signals next to the signal of interest at 11ppm. The legend of Fig S4C mentions 'intense peaks are from phenol', however only the 4 signals above 100ppm arise from phenol. No explanation is given for signals between 50ppm and 100ppm, no full reference spectra are added in conditions without dUMP or ThyX.

'Our structure is perfectly superimposable with that of the ThyX/dUMP/folate ternary complex previously reported (r105)'

This statement is hard to judge without having access to the pdb or RMSD values. Fig 3c shows a model of ThyX in complex with the flavin carbinolamine, dUMP and folate that was obtained by superimposing the crystal structure of FADH- 308 •CH₂O complex with the structure of ThyX in complex with dUMP and folate (pdb, 4gt9). It is impossible to see any steric clashes since only the ribbon diagram of ThyX (determined in this work?) is shown.

The legend of fig 3 mentions R174 participating in activation of dUMP, though no reference is given for this statement. Authors argue that 'The structure shows that Tyr91 and Ser88 (Fig 3d), previously shown to be critical for activity by site-directed mutagenesis, could stabilize such a conformation via H-bonding' also here a reference is missing. The legend of fig3d mentions 'This model shows the different orientations adopted by these residues in two different structures of ThyX' though it does not mention structures that are used for this overlay.

Reviewer #3 (Remarks to the Author):

Bou-Nader et al. interrogate the mechanism of an alternative, flavin-dependent thymidylate synthase (FDTS), with biochemical and structural biology methods. FDTS is an attractive antimicrobial target because it isn't found in humans and employs chemistry orthogonal to that of the human enzyme. Using formaldehyde as a proxy for the biological carbon donor, methylenetetrahydrofolate (MTHF), the authors beautifully demonstrate that carbon travels to its final acceptor (dUMP) via a novel N5 flavin carbinolamine intermediate. This intermediate is observed in *cristallo* and spectroscopically and is shown to be catalytically competent. The work is elegantly performed, with necessary controls, and will be of significant interest not only to the drug-development community, but also to the flavoenzyme community, adding to the growing collection of covalent flavin adducts as reactive species.

One suggestion I have is to make it clearer in the text that formaldehyde is used as a tool here, to avoid potential confusion that authors are claiming formaldehyde as the biological carbon donor *in vivo*. For instance, the authors might compare K_d values for MTHF (tens of μM) vs formaldehyde (tens of mM) and state that large discrepancy agrees with folate serving as a CH₂O carrier/positioner, thanks to its specific interactions with ThyX and its flavin.

Minor comment:

Fig. S4: The spectra are very small and difficult to read. Can you please enlarge each spectrum and label peaks of interest, specifically the ~12 ppm peak for dTMP's methyl group in panel c?

REVIEWER COMMENTS

Reviewer #1 (Remarks to the Author):

This manuscript proposed the chemical mechanism of the methylation by ThyX to produce the thymidine nucleotide. Although the tertiary structures of ThyX from several organisms have been determined so far, the reaction mechanism has been unclear. The authors demonstrated the presence of the carbinolamine intermediate by the X-ray crystallography. Also, the authors showed that formaldehyde can be used instead of methylene-THF to produce dTMP from dUMP. Based on these results, a novel chemical mechanism for ThyX is proposed. The elucidation of chemical mechanism is fundamental in biochemistry and this manuscript gave new insight of the enzymatic reaction. However, the following points will be addressed by the authors to justify the proposed mechanism.

We thank the reviewer for his/her positive comments.

1. In the proposed chemical mechanism by Mishanina et al. 2016, the methylene group was transferred to flavin to form iminium intermediate (CH₂-flavin) instead of carbinolamine intermediate (CHO-flavin). Indeed, the S_N2 reaction between the C5 of dUMP and the methylene carbon of carbinolamine intermediate is reasonable. Still, it is possible that the carbinolamine intermediate is formed via iminium intermediate. Furthermore, it is also possible that iminium form is the active intermediate and the iminium intermediate is formed from carbinolamine intermediate in the CH₂O-shunt pathway. It is recommended to discuss on these possibilities.

The reviewer is absolutely right. With formaldehyde, we necessarily have the formation of a flavin-carbinolamine species, which can subsequently evolve to its iminium counterpart. In the case of the reaction with CH₂THF, the flavin-iminium is believed to form first via an addition-elimination reaction between the reduced flavin and folate. Likewise, this iminium can subsequently give the carbinolamine.

Of course, the previously proposed flavin-iminium electrophile could be the *bona fide* methylating agent. However, here we provide further pieces of evidence that we believe are in disfavor of a flavin-iminium. Studies by Kemal & Bruice in 1976 (see in references of the revised article) showed that the solvolysis of a model flavin generated the carbinolamine likely formed by the rapid reaction of the iminium with water, attesting to

the extreme instability of the iminium. As a matter of fact, ThyX presents a large active site that is quite exposed to the solvent and which should prevent formation and reaction of the iminium with dUMP. It is therefore expected that formation of the carbinolamine species from the iminium counterpart occurs at a much faster rate than that of the carbon transfer reaction to dUMP. Furthermore the sp^3 -hybridized carbinolamine presents a favorable distance and geometry for the in-line attack of the nucleophile in carbon transfer and water displacement. This is compared to the geometric requirements a nucleophilic attack on an sp^2 -hybridized iminium (Fig S14). Efficient attack of nucleophiles on the π -system of carbonyls or imines occurs along the so-called Bürgi-Dunitz trajectory JACS (1973) 95, 5065-5067, with the nucleophile attacking the unsaturated carbon at vector $\sim 107^\circ$ from the C-N or C-O bond in a direction towards the electronegative atom (Light et al 2014). If water were eliminated from the carbinolamine-flavin adduct we observed crystallographically, the methylene carbon of the planar iminium-flavin adduct would be positioned over the C-N double-bond and much too distant to attack the iminium carbon, the C-N vector of the iminium would be almost 180° away from the direction needed for attack.

Here a stereo view showing ThyX active site with the dUMP and the flavin-carbinolamine and in which the OH of the carbinolamine adduct is rotated to its reactive position (either for displacement or elimination). It's obvious from the figure that elimination to form the iminium puts the CH₂ in a place where it simply cannot react; it would take more than just a little dynamical changes to achieve a reactive conformation. In obvious contrast, the carbinolamine is poised to react.

Therefore, these stereoelectronic considerations lead us to favor the carbinolamine as the actual carbon-transfer agent. We have discussed these issues in the revised manuscript.

2. Catalytic mechanism is also important. Because crystal structures were reported in the manuscript, the authors may discuss on the contribution of amino acid residues including Tyr91 and Ser88 to the catalytic mechanism of ThyX. For example, how the H₂O molecule is generated during the reaction?

Although Pka of these residues are not available in ThyX, serine and tyrosine could act as acid by protonating the carbinolamine. Alternatively, these residues could polarize a water molecule into the active site of ThyX, which would serve as a proton donor. However the fact that phenolic oxygen of Tyr 91 is within hydrogen-bonding distance to a guanidinium nitrogen of Arg 90, which is also conserved among ThyX homologues, leads us to favor the first scenario.

We now further elaborate on the catalytic mechanism by stating: “The carbinolamine geometry favors an SN2-like attack by dUMP, which is activated as a nucleophile by deprotonation of N3 through electrostatic interaction with R174, enhancing the enamine-character of the pyrimidine moiety. Nucleophilic attack leads to the formation of a covalent FAD-CH2-dUMP adduct, previously proposed based upon the fragments obtained from rapid-quenching by base²⁷, and the displacement of a water molecule. The two conserved residues, Ser88 and Tyr91, at hydrogen bonding distance with the carbinolamine hydroxyl group may assist the SN2 reaction by acting as acids and promoting the displacement of the β -hydroxyl leaving group (Fig 3d). It is worth noting that the phenolic oxygen of Tyr 91 is within hydrogen-bonding distance to a guanidinium nitrogen of Arg 90 (also conserved); the proximity of such a (presumably) positive charge could enhance the acidity of Tyr 91, which might be the general acid catalyst that assists the displacement of the leaving water.”

Furthermore, it is helpful for general readers to show the conservation of these amino acid residues in sequences as well as structures.

Sequence alignment showing the strict conservation of S88 and Y91 among ThyX enzymes has been added as FigS13.

Reviewer #2 (Remarks to the Author):

Authors demonstrate that in anaerobic conditions formaldehyde can replace the natural methylene donor CH₂THF in ThyX catalyzed methylation of dUMP. These findings are highly interesting and endorsed by crystallographic, NMR and HPLC data. However, comparison of reaction kinetics with the native methylene donor should be included.

**We thank the reviewer for his/her positive feedback.
A kinetic trace with the CH₂THF has been added in the new figure 2A.**

In the crystal structure a the carbinolamine adduct of reduced FADH- is observed. Authors are convinced this is the methylating agent and not a trapped intermediate that should be activated into a reactive imine since 'The methylene group of the carbinolamine adduct is nicely located only 2.4 Å from C5-dUMP and is ideally poised for an SN₂ attack by the activated nucleobase to generate an intermolecular C-C bond.' However, without MD these are not convincing arguments. An MD is the least that should be performed on the obtained model to have a more reliable idea on distances and hydrogen bonding interactions indicated in fig 3d, to endorse the hypothesis that carbinolamine flavin species could be the active species instead of its iminium counterpart, that is so far postulated as the dUMP methylating agent. For comparison: To date, the 5-HOCH₂H₄folate that was considered as structural evidence for the 5-iminium ion intermediate, which is the proposed reactive form of CH₂H₄folate in ThyA (Biochemistry. 1993 Jul 20;32(28):7116-25. doi: 10.1021/bi00079a007) and also the 5-hydroxymethyl-dUMP was identified as an acid trapped intermediate in the reaction catalyzed by ThyX (T.V. Mishanina, et al. Trapping of an intermediate in the reaction catalyzed by flavin-dependent thymidylate synthase J. Am. Chem. Soc., 134 (9) (2012), pp. 4442-4448).

We thank the reviewer for his proposal regarding the molecular dynamics. However, the complexity of the system, a tetrameric protein with 4 active sites, each of them containing a flavin-carbinolamine, dUMP and folate, makes parametrization very complicated. We fully agree that this may be a direction to explore in the future and it is a very demanding study of investigations in itself. In the current state, the structure seems to us to be fairly conclusive on the fact of the presence of a reaction intermediate, which acts as a methylating agent for dUMP. However, we here provide further pieces of evidence that we believe are in disfavor of a flavin-iminium. Studies by Kemal & Bruice in 1976 (see in references of the revised article) showed that the solvolysis of a model flavin generated the carbinolamine likely formed by the rapid reaction of the iminium with water, attesting to the extreme instability of the iminium. As a matter of fact, ThyX presents a large active site that is quite exposed to the solvent and which should prevent formation and reaction of the iminium with dUMP. It is therefore expected that formation of the carbinolamine species from the iminium counterpart occurs at a much faster rate than that of the carbon transfer reaction to dUMP. Furthermore the sp³-hybridized carbinolamine presents a favorable distance and geometry for the in-line attack of the nucleophile in carbon transfer and water displacement. This is compared to the geometric requirements a nucleophilic attack on an sp²-hybridized iminium (Fig S14). Efficient attack of nucleophiles on the π-system of carbonyls or imines occurs along the so-called Bürgi-Dunitz trajectory JACS (1973) 95, 5065-5067, with the nucleophile attacking the unsaturated carbon at vector ~107° from the C-N or C-O bond in a direction towards the electronegative atom (Light et al 2014). If water were eliminated from the carbinolamine-flavin adduct we observed crystallographically, the methylene carbon of the planar iminium-flavin adduct would be positioned over the C-N double-bond and much too distant to attack the iminium carbon,

the C-N vector of the iminium would be almost 180° away from the direction needed for attack.

Here a stereo view showing ThyX active site with the dUMP and the flavin-carbinolamine and in which the OH of the carbinolamine adduct is rotated to its reactive position (either for displacement or elimination). It's obvious from the figure that elimination to form the iminium puts the CH₂ in a place where it simply cannot react; it would take more than just a little dynamical changes to achieve a reactive conformation. In obvious contrast, the carbinolamine is poised to react.

Therefore, these stereoelectronic considerations lead us to favor the carbinolamine as the actual carbon-transfer agent. We have discussed these issues in the revised manuscript.

The top spectrum in fig 2A is an expansion of the full NMR spectrum (not RMN!) in Fig S4C. The full spectrum clearly shows many signals next to the signal of interest at 11ppm. The legend of Fig S4C mentions 'intense peaks are from phenol', however only the 4 signals above 100ppm arise from phenol. No explanation is given for signals between 50ppm and 100ppm, no full reference spectra are added in conditions without dUMP or ThyX.

We corrected the term NMR. We also corrected our typo error and replaced the term dUMP by dTMP in the legend of figure S4A. We also added figure S4D (reference spectra without ThyX), which does not have the signal of interest at 11 ppm. As the reviewer mentioned, the peaks above 100 ppm (at 115, 121, 129 and 155 ppm) arise from phenol. As for the other peaks between 50 and 100 ppm, the peak at 81 ppm arises from formaldehyde as shown in the NMR spectrum of formaldehyde (Figure S4B), the peak at 62 ppm probably arises from Tris-Cl buffer and is present in activity tests without ThyX (Figure S4D). The other minor peaks at 59, 72, 87 and 88 ppm could not be attributed but are also present in activity tests without ThyX (Figure S4D;) indicating that they are not arising from an activity of ThyX. These peaks could arise from impurities in phenol used for products extraction.

'Our structure is perfectly superimposable with that of the ThyX/dUMP/folate ternary complex previously reported (r105)'

This statement is hard to judge without having access to the pdb or RMSD values.

We now state in the result section "The structures show that the homotetrameric enzyme does not undergo significant conformational changes with overall root-mean-square deviation (RMSD) of 0.2 Å over 167 residues".

Fig 3c shows a model of ThyX in complex with the flavin carbinolamine, dUMP and folate that was obtained by superimposing the crystal structure of FADH- 308 •CH₂O complex with the structure of ThyX in complex with dUMP and folate (pdb, 4gt9). It is impossible to see any steric clashes since only the ribbon diagram of ThyX (determined in this work?) is shown.

Our initial Fig S8 shows the structural superposition of our structures with the previously reported structure of ThyX bound to CH₂THF and dUMP (PDB 4gt9). We have expanded this by plotting the RMSD for each residue of each subunit. This further confirms that minor rearrangements of some side chains occur while the overall structures are identical.

Additionally, we have added a figure in the main manuscript (new Fig S12) showing a stereo view of all the side chains in the active site at the vicinity of the flavin-carbinolamine in our structures in the presence of dUMP and CH₂THF (pdb 4gt9). There is only the CH₂OH moiety on the N5 of flavin that clashes with dUMP.

The legend of fig 3 mentions R174 participating in activation of dUMP, though no reference is given for this statement.

The reference DOI 10.1021/bi500648n has been added.

Authors argue that 'The structure shows that Tyr91 and Ser88 (Fig 3d), previously shown to be critical for activity by site-directed mutagenesis, could stabilize such a conformation via H-bonding' also here a reference is missing.

The reference DOI 10.1021/bi500648n has been added.

The legend of fig3d mentions 'This model shows the different orientations adopted by these residues in two different structures of ThyX' though it does not mention structures that are used for this overlay.

The structures used for this overlay are: (i) ThyX-FADH⁺ soaked with 20 mM Formaldehyde vs (ii) pdb 4gt9

Reviewer #3 (Remarks to the Author):

Bou-Nader et al. interrogate the mechanism of an alternative, flavin-dependent thymidylate synthase (FDTS), with biochemical and structural biology methods. FDTS is an attractive antimicrobial target because it isn't found in humans and employs chemistry orthogonal to that of the human enzyme. Using formaldehyde as a proxy for the biological carbon donor, methylenetetrahydrofolate (MTHF), the authors beautifully demonstrate that carbon travels to its final acceptor (dUMP) via a novel N5 flavin carbinolamine intermediate. This intermediate is observed in *cristallo* and spectroscopically and is shown to be catalytically competent. The work is elegantly performed, with necessary controls, and will be of significant interest not only to the drug-development community, but also to the flavoenzyme community, adding to the growing collection of covalent flavin adducts as reactive species.

We thank the reviewer for his/her positive feedback.

One suggestion I have is to make it clearer in the text that formaldehyde is used as a tool here, to avoid potential confusion that authors are claiming formaldehyde as the biological carbon donor *in vivo*. For instance, the authors might compare K_d values for MTHF (tens of μM) vs formaldehyde (tens of mM) and state that large discrepancy agrees with folate serving as a CH_2O carrier/positioner, thanks to its specific interactions with ThyX and its flavin.

To make it clearer that formaldehyde is used as a tool in our study we have added the following statements:

In the introduction we now say, “we show that a CH_2O -shunt can replace the natural methylene donor for ThyX-dependent dUMP methylation”.

In the result section we emphasized this by stating “Taken together, these results confirmed that ThyX uses CH_2O as a direct methylene donor for dTMP synthesis and can therefore substitute for CH_2THF by forming a flavin intermediate that could be a carbinolamine adduct.”

In the discussion we have added: “The tighter binding of ThyX for CH_2THF ($K_D \sim 4\mu\text{M}^{27}$) compared to CH_2O ($K_D \sim 20\text{ mM}$) confirms that methylene tetrahydrofolate acts as the biological carbon donor for ThyX, serving as a CH_2O carrier/transporter and thus avoiding genotoxic effects²⁸⁻³⁰.”

Minor comment:

Fig. S4: The spectra are very small and difficult to read. Can you please enlarge each spectrum and label peaks of interest, specifically the ~ 12 ppm peak for dTMP's methyl group in panel c?

We enlarged all NMR spectra and added a new figure (S4D), which is a reference spectrum without ThyX.

REVIEWER COMMENTS

Reviewer #1 (Remarks to the Author):

This revision addressed all my comments.

Reviewer #2 (Remarks to the Author):

taking into account most of the comments of reviewers, the revised manuscript is significantly improved compared to its previous version. Therefore I recommend for publication.